# Lessons from COVID’S Vaccination: External-Internal Frictions and Efficiency

**DOI:** 10.3390/vaccines11020248

**Published:** 2023-01-22

**Authors:** Aldo Ramirez-Zamudio, Elmer Sanchez Davila

**Affiliations:** 1Department of Economics, Universidad de Lima, Av. Javier Prado Este 4600, Lima 15023, Peru; 2Department of Economics, University of Talca, Av. Lircay S/N, Talca 3460000, Chile

**Keywords:** vaccines, covid, variants

## Abstract

This paper explores some economic variables that determine the emerging of new COVID-19 variants and the determinants of vaccination advances in 108 countries during a quarterly period from March 2020 to March 2022. We found that more people being fully vaccinated and more education (measured as schooling years) decrease the probability of the emergence of new COVID-19 variants, but more crowded cities and higher percentages of urban population increase that probability. Furthermore, we found that the percentage of fully vaccinated people depends positively on the country’s preparation to respond to a health crisis, educational levels, and the index of economic complexity (which measures how diverse in the production of goods and services a country is and the level of its infrastructure), and it depends negatively on the percentage of rural populations (which makes vaccination more difficult).

## 1. Introduction

Since the appearance of COVID-19 in November 2019, the world has been severely affected in many fronts. The global economy fell 3.1% in 2020, according to the International Monetary Fund [1], being the largest drop since 1980; also, in August 2022, the World Health Organization (WHO) estimated COVID-19-related deaths to be about 6.5 million [2].

Responding to the severity of the virus, the main laboratories of the world, supported by various countries, started immediately the search for an effective vaccine. By the end of 2020, at least three public-private initiatives had managed to pass stage III of research and managed to request authorization from health regulators to commercialize them. In December 2020, the vaccination process began in some countries, especially those that had directly invested in the development of the vaccines.

Initially, we could observe an unequal access to vaccines; moreover, most high-income countries received vaccines in numbers that surpassed largely their entire populations. This initial inequality in the access to vaccines caused WHO to coin the health crisis as a “two-way pandemic”, with high-income countries receiving 75% of the available vaccines and the majority of the countries having only the remaining 25%”. Then, with the firm aim of putting an end to the pandemic in 2022, WHO’s general director insisted on the need of eliminating the inequity in the vaccination process, assuring that populations will be entirely vaccinated by the middle of this year. However, by observing the vaccination rates around the world, this appears to be yet far from becoming true.

Indeed, according to Our World in Data [3], until September 2022, only 69.7% of the world’s population received at least one dose of a COVID-19 vaccine. In addition, only Latin America (81%), Asia-Pacific (80%), United States and Canada (80%), and Europe (69%) have high complete-vaccination rates above the global average (66%), while the Middle East and Africa could only reach a vaccination rate of 58% and 28%, respectively.

Likewise, by September 2022, Chile and Peru had the highest vaccination rates (93% and 92%, respectively) not only in Latin America but also worldwide, which could be achieved due to the necessary financial resources to acquire the vaccines and an efficient vaccine distribution strategy. On the contrary, the countries with the worst performances were mainly African countries such as Congo (4.5%), Madagascar (5.4%), Cameroon (6.1%), Senegal (9%), Mali (11%), etc. [3].

Therefore, the pandemic is still a serious health problem around the world, and it appears that it will be around for long. We are not sure yet if the virus will turn into less aggressive forms becoming endemic, requiring less effort from our health systems and fewer restrictions to people’s mobility or, on the other hand, it will continue mutating and eventually to more aggressive versions, demanding immediate action by improving current vaccines, health infrastructure, prevention or developing new treatments.

Public authorities around the world have insisted in vaccination as being the most effective way to face the virus. Moreover, they have exerted major efforts to acquire and inoculate vaccines to their citizens. However, the willingness of people in accepting vaccination can also alter the herd immunity [4] and there still exist large asymmetries in the number of completely vaccinated people in different countries. At the same time, the effort associated to the creation of vaccines in such a record time has been, without doubts, a direct consequence of private laboratories’ research, partially supported by governments [5]. It is very likely that these rapid results could not have been obtained without patents and intellectual protection rights and laws [6]. Both pharmaceutical companies (which operate as oligopolies) and patents have carried quick advances in obtaining the vaccines; however, they also imply bigger proportional costs (on GDP) to less developed countries and, according to some recent economic literature [6,7], should been better defined, since they can slow or even impede the development of new products, or generate indefinite socially inefficient positions of domain [8].

Even though manufacturing vaccines in sufficient quantities for the enormous demand and their subsequent distribution were noticeable issues at the beginning, problems in the global supply chain, the lack of infrastructure necessary to store and preserve vaccines, as well as the bureaucratic frictions typical of many countries, such as their logistical inability to be able to quickly inoculate their target population, could have caused the virus to cause many more deaths and to mutate to new variants. Even first-generation vaccines may have lost effectiveness against new variants [9]. This fact affects not only those countries with lower inoculation rates but the whole world, due to the large number of people who move between countries and continents, which can spread the new variants, in very short times and put the whole world in serious danger again.

So, what are the observable economic factors that influence both the appearance of new variants of the coronavirus and the advances in the vaccination rate in 108 countries in a quarterly period from March 2020 to March 2022? These are the main questions this work intends to answer. We know that a war between Ukraine and Russia started in the first half-year of 2022; however, the effects on main variables are marginal, since our period of study overlaps with the war’s period only in one month.

Therefore, one of the objectives of this work is to find some observable variables determining the appearance of new variants of the virus, and if the variable “fully vaccinated people” results significative (as it is in our results), it may demonstrate that unequal access to vaccines can be an important issue to be accounted for all nations.

The other objective is to try to find some country-related factors influencing the process of vaccination for 108 countries (those with available data) which may be valuable to correct, or a reason to implement, public policies aimed to improve our response to this and future pandemics. In general, this work may help to recall politicians, authorities, and policymakers that global vaccination should advance more homogeneously both along countries and in the velocity of inoculation, which would reduce deaths and the probability of the occurrence of new mutations of the virus that could end up affecting the whole world.

In line with the objectives, the following specific hypotheses were raised:(1)High vaccination rates and high levels of education [10] decrease the probability of the appearance of new COVID-19 mutations, whereas population overcrowding and large urban populations cause the opposite effect.(2)Improved health disaster preparedness, urban concentration, and relatively high economic complexity are expected to be associated with high coronavirus vaccination rates. On the other hand, large rural populations are inversely related with vaccination advances.

If our hypotheses can be demonstrated, this work would contribute to determine that the release of patents demanded by some countries to the WTO is not a sufficient condition to end the virus and that there are other important factors from the economic point of view that we must also observe, such as the capacities and dynamics of countries to acquire, distribute, and inoculate vaccines.

On this vein, it appears to be clear that we should quickly vaccinate as many people as we can, but we must know whether the access to vaccines per se is sufficient or whether we should also identify other problems or bottlenecks that require the same or more attention. In relation to this point, even international collaboration could be more efficient and faster than pushing for the revision of international treaties on intellectual property protection such as the Trade-Related Aspects of Intellectual Property Rights (TRIPS) agreements of 1995, which may entail discussions and negotiations which would go on too long considering the urgency of the problem.

## 2. Related Literature

About socio-economic variables affecting vaccination, Sarkar and Morshed examined Bangladesh in 2021 and found that both demographic factors (population density and urban population) and economic factors (percentage of industrial workers) are keys to the spatial priority of vaccine implementation [11]. Also, Ambros and Frenkel investigated the same phenomena in Germany in November 2021 and found that the COVID-19 mortality rate and population density were associated with an increase in the vaccination rate, while the percentage of the adult population generated the opposite effect [12]. Later, Gertz et al. examined 843,985 surveys in the US between February and November 2021, finding that higher-income people were more likely to complete vaccinations than those with lower income [13]. Then, Lee and Huang evaluated Nueces County in Texas and found that people from low socioeconomic strata have low vaccination rates; likewise, the rejection of the vaccine in certain communities has spread to nearby communities of a similar socioeconomic position [14].

Furthermore, Agarwal et al. examined 756 US counties, where the variables related to factors of socioeconomic privilege and political ideology influenced racial disparity in the vaccination rate [15]. Also, Roghani used data from 25 countries from February to August 2021, finding that higher Gross Domestic Product (GDP) per capita was positively associated with greater vaccine acquisitions [16]. Finally, Irfan et al. examined surveys of 754 households in Pakistan and found that pandemic risk perceptions positively affected the vaccination rate, while the cost to access the vaccine (transportation and leaving current jobs), and its unavailability, had the opposite effect [17].

In terms of people’s willingness to vaccinate, Urrunaga-Pastor et al. used 784,460 surveys of people from Latin American Countries (LAC) and find that living in a rural area, economic insecurity, and having depressive symptoms were associated with a higher probability of fearing the adverse effects of vaccines [18]. Also, Lazarus et al. used a sample of 13,426 people from 19 countries during a high rate of spread of COVID-19 in June 2020 and found that the heterogeneity by demographic factors of those surveyed to accept a vaccine (when it would be available) was substantial in explaining later the vaccination rate. They showed too that people with high levels of trust in a government’s information were more likely to accept a vaccine [19,20]. Cerda and García used 370 surveys in Chile and found that the variables related to the information on the severity of the vaccine side effects and effectiveness mainly explained the possibility of a vaccine rejection [21]. Khaled et al. studied Qatar between December 2020 and January 2021 and found that a lack of concern about catching the virus was associated with resistance to the vaccine, mainly in female, Arab ethnic groups and non-immigrant Qataris [22]. Khan et al. investigated 17 countries between November 2020 and April 2021 and found that people who were more exposed to warnings against the vaccine are less likely to be vaccinated, which reflects the importance of information management [23]. Also, Ren et al. investigated 416 subjects in China in 2021 and obtained a high rate of vaccination of patients, to the extent that it provided them with specific information on the safety and importance of the vaccine [24].

Harper et al. analyzed Facebook vaccine-related discussions in Australia from December 2020 and February 2022, they found that controlling access to or censoring vaccine-critical misinformation did not reduce prejudices and negative beliefs about vaccines but even reinforced them [25]. Moreover, discussions deviate to political and social arguments. Similar results are obtained by Kwanho et al. who found ample misinformation about COVID-19 vaccines even in the public media. They argued that exposure to misinformation increases the perceived risk of getting a vaccine and even evokes negative emotions towards them, reducing vaccination intentions [26]. Furthermore, Viswanath et al. used surveys of 1012 representative adults in the US and found that a higher perception of the risk of contagion and less schooling were positively and negatively associated with vaccine acceptance, respectively [27]. Adedeji-Adenola et al. used 1058 adult subjects in Nigeria from April to June 2021 and found that a high level of awareness was positively associated with higher vaccination rates, but this relationship may be affected by variables such as religion or occupation [28]. Davis et al. investigated six low- and middle-income countries and found that perceived social norms, perceived positive and negative consequences, perceived risk, and access to vaccines had the highest associations with COVID-19 vaccine acceptance [29]. Holzmann-Litting et al. investigated 4500 health care workers in Germany in February 2021 and found a significant relationship between refusing COVID-19 vaccinations and the fear of side effects [30]. In the same line, Huang et al. studied surveys of 1047 primary care professionals in the US at the beginning of 2021 and found that greater confidence in the COVID-19 vaccine, perceiving more positive social norms, and receiving recommendations to be vaccinated were associated with greater acceptance of the vaccine [31]. Finally, Salman et al. used a survey of six countries (Pakistan, Saudi Arabia, India, Malaysia, Sudan, and Egypt) from April to August 2022 and found a significant positive correlation between conspiracy beliefs and vaccine hesitancy [32]. All these works argue for the importance of understanding the diverse factors that impact the effectiveness of communication–including the context in which it is received–and the emergent properties created by communication processes.

On the other hand, at the time of writing this work, we have not found any research on economic variables determining the appearance of new COVID-19 variants or economic country-related variables (like we use) explaining vaccination rates, which will be our main interest from now.

## 3. Methodology

### 3.1. Variables and Data

As we mentioned before, the paper has two main objectives: (i) to find some observable variables that determine the appearance of new variants of the virus, and (ii) to find some country-related factors that influence the process of vaccination for 108 countries.

The sample in both models used the largest amount of data available. We used a quarterly frequency panel dataset of 108 countries from the first quarter of 2020 to the first quarter of 2022. The panel data estimated is considered as a micro panel, given that the number of cross sections (108) is greater than the number of time periods (9), and it is a strongly balanced panel given that there are not missing values. In total, the panel contains 972 observations, and the data was obtained from different worldwide known international organization sources, such as: World Health Organization (WHO), United Nations Development Program (UNDP), World Bank (WB), and some others.

### 3.2. Econometric Methodology

The first model is estimated by a logit-panel model given that our endogenous variable, COVID-19 variants (Y_i), is a dummy variable in which 1 refers to the appearance of a new relevant COVID-19 variant, and 0 otherwise. Moreover, the methodology of a Logit-Panel model fits when there are some variables that change and others maintain the same values for long periods, such as education, overcrowding, or geographic diversity (see Table 1).

Then, to verify the determinants of the appearance of new variants of the virus, we propose a logit-panel data model which allows us to estimate the probability of the occurrence of an event under certain circumstances. Also, the main characteristic of this probability model is that the endogenous variable of the model is a dichotomous variable, so its estimation is carried out by maximum likelihood. In addition, this probability model may be of fixed or random effects [33,34] which are expressed as follows:Fixed effects: Y_it = β_i + β_k × X_kit + ε_i(1)
Random effects: Y_it = β_0 + β_i × X_it + μ_it(2)
where Y_it represents a dummy variable which refers to the appearance of new COVID-19 variants, X_it represents the vector of exogenous variables in the model and ε_it and μ it are the error terms of the estimation for the first and second model, respectively.

To determine which kind of effects we should use, we first estimated both effects and then used the Hausman test (see Appendix A) to decide if our panel data behaved better under a fixed or a random effect assumption. Finally, we can obtain the average partial effects to show marginal effects and the probabilities associated with the independent variables in the occurrence probability of the appearance of a new COVID-19 variant.

Also, to obtain consistent estimators using a binary dependent variable [35], we considered a kind of binary response model of the form:P(y_it = 1│x_it, a) = G(a + β_1*x_1it + β_2*x_2it + β_3*x_3it + β_4*x_4it) = G(a + xβ) (3)
where G is a function that assumes values strictly between zero and one: 0 < G(z) < 1, for all the real numbers z [36]. The regressors (X_i) presented in Table 1 are widely used in the health economics literature, so they are significant to explain the endogenous variable (the appearance or non-appearance of new variants). In this sense, the importance of this model is understood; for example, if it is shown that overcrowding contributes to the appearance of new variants of the virus, it is possible to ensure that targeted quarantines policies could be effective to reduce the appearance of a new relevant COVID-19 variant.

The second model is estimated also by a panel data model. We also used a Hausman test to decide which kind of effects behave better under our data (see Appendix B). Although the results of the Hausman test exposed that we should use a fixed effects model, we decided to apply a random effects panel data model because we are interested in knowing the evolution of vaccination rates in relation to the exogeneous variables (see Table 2).

Panel data estimations have three recurring problems. First, the generalized least squares estimators with autocorrelation do not show the highest maximum likelihood estimators. Therefore, it is crucial to define whether or not the model presents serial autocorrelation. Then we used the Baltagi-Li test, where its null hypothesis is that there is no first-order autocorrelation. Second, it is necessary to know if the variance of the errors of each unit of analysis is constant or not. Hence, we used the Breusch–Pagan test because it is not sensitive to the normality of the errors. Third, the estimates may have contemporary correlation problems, which is a phenomenon where any observation affects the others. To detect this problem, we used Pesaran’s test, where the null hypothesis is that there is cross-sectional independence, or that the errors between the units are independent.

In Appendix C, Appendix D and Appendix E, it is shown that the model has problems such as serial correlation, contemporary correlation, and heteroskedasticity. Therefore, in order to solve these three problems, we used linear regression with correlated panels.

It is important to mention that other variables are also important for the model; however, due to the lack of data, it is impossible to include them as our regressors, since we assume they are not significant variables, they would be part of our error term without causing any biased results.

## 4. Empirical Results and Analysis

### 4.1. First Model: New Variants of the COVID-19

To study which observable variables determine the appearance of new variants of the virus, we focused on the following variants: Dseta, Gamma, Mu, Delta and Kappa, Lambda, Zeta, Beta, Omicron, Alpha, Eta, Epsilon, and Iota. Table 3 contains a summary of our exogenous variables where the main strains of COVID-19 emerged. A common feature of the new variants of COVID-19 is that it emerges when the percentage of people fully vaccinated is almost non-existent.

In Table 3, we show the values of the independent variables of the countries where the COVID-19 variants initially emerged, which are compared with the data of the same variables in a sample of countries weighed by their population at the moment that the respective COVID-19 variant appeared. The countries reported in Table 3 are the ones where there is evidence of a new COVID-19 variant. It is known that China suffered from a COVID-19 variant; however, its data is not available. Some other countries, such as Philippines, did suffer from a strain of a COVID-19. One of our objectives is to find some observable variables that determine the appearance of new variants of the virus, not strains of the virus. For instance, when the Omicron variant appeared in South Africa, around 26% of the population were fully vaccinated, while in the rest of the countries, this number was around 49%.

Table 4 shows the estimations of our model using a panel data with random effects. As we can see, variables such as the percentage of people fully vaccinated, crowded countries (cities with more than 10 million residents), and urbanization explains significatively the appearance of a new COVID-19 variant.

Table 5 shows the marginal effects of our panel data model. For instance, we can conclude that countries which have more fully vaccinated people are related with 7.0% less probability that a new COVID-19 variant appears compared to countries that have low vaccination rates. This result happens because if the population that has been inoculated with the first and second doses increases, it is less probable that a person infected with the coronavirus could relapse and, hence, allow the virus to mutate. Moreover, crowded countries also impact significatively on the probability that a new strain of COVID-19 emerges. Living in a country with more crowded cities is related to a 2.0% higher probability that a new COVID-19 variant appears, compared to countries with less or no crowded cities. Furthermore, the nations which have a high urban population have a 0.05% higher probability that a new COVID-19 variant appears. Also, countries that have a population with high years of schooling are associated with 0.17% less of a probability that a new COVID-19 variant appears, compared to countries that have a population with lower years of schooling. However, this variable seems to be not significant to explain new COVID-19 strains.

### 4.2. Second Model: Determinants of the Vaccination Process

To find some country-related factors influencing the process of vaccination around our cross-section data, we estimate a panel data with random effects. It seems that all our exogenous variables might be significant to explain the process of vaccination, the most important variable to explain our endogenous variable concerns the preparedness and response status of a country.

Table 6 shows that Africa and the rest of Asia have, on average, the highest rural population percentages with respect to the sample. In addition, the region of Central America has the lowest value of the preparedness and response status, which is below the average sample together with Africa and the rest of Asia. Furthermore, North America, Europe, and the rich countries of Asia have the highest average years of schooling in relation to the sample average. Otherwise, Central America, South America, Africa, the rest of Asia, and the rest of the countries have lower values of the economic complexity index.

According to the results of Table 7, being prepared for a sanitary disaster has a positive impact on increasing the vaccination rate. This result happens because the definition of being prepared for a sanitary disaster entails the pandemic influenza preparedness plans and country readiness assessment for health emergencies; so, the more prepared the country is, the more likely it is to have higher increasing vaccination rates. Education has also a positive and significant impact on increasing the vaccination rate. The more educated people are, the more likely they would understand the benefits of vaccination to prevent new contagions.

On the other hand, higher rural populations have a negative impact on the vaccination rate. This is obvious, as it is more complicated for a country to vaccinate its population if there is a lack of infrastructure and logistics, and if the population is more spread out from urban zones. The withing country migration will be interesting to analyze; however, the lack of data makes it impossible. Nonetheless, Suresh et al. analyzed this factor in India [37]. Finally, the index of economic complexity is related to vaccination rates where countries with a higher index are usually developed countries, which have more power to get vaccines faster than less developed countries. In general, the results of both models recall politicians, authorities, and policymakers that global vaccination should advance more homogeneously, given that it would reduce deaths and the probability of occurrence of new mutations of the virus that end up affecting the whole world.

## Figures and Tables

**Table 1 vaccines-11-00248-t001:** Variable description for the first model.

Variables	Description	Source
COVID-19 variants (Y_i)	1 (The appearance of a new relevant COVID-19 variant before the population reaches a 50% vaccination rate of the first and second doses) and 0 (Otherwise).	World Health Organization (https://www.who.int/es/activities/tracking-SARS-CoV-2-variants (accessed on 4 July 2022)
Percentage of people fully vaccinated (x_1)	The total number of people who received complete doses (first and second) prescribed by the initial vaccination protocol divided by the total population.	Our World in Data (https://ourworldindata.org/grapher/share-people-fully-vaccinated-covid (accessed on 4 July 2022)
Years of schooling (x_2)	Mean years of total schooling across all education levels	United Nations Development Program (https://hdr.undp.org/data-center/documentation-and-downloads (accessed on 4 July 2022)
Crowding (x_3)	The number of cities having more than ten million people	NGIA, US Geological Survey, US Census Bureau, and NASA (https://simplemaps.com/data/world-cities (accessed on 4 July 2022)
Urban population (x_4)	It refers to the percentage of people who lives in urban areas	World Bank (https://data.worldbank.org/indicator/SP.URB.TOTL.IN.ZS (accessed on 4 July 2022)

**Table 2 vaccines-11-00248-t002:** Variable description for the second model.

Variables	Description	Source
Percentage of people fully vaccinated (Y_i)	The total number of people who received complete doses (first and second) prescribed by the initial vaccination protocol divided by the total population.	Our World in Data (https://ourworldindata.org/grapher/share-people-fully-vaccinated-covid (accessed on 4 July 2022))
Operational readiness index (x_1)	An indicator based on additional information from voluntary joint external evaluations, pandemic influenza preparedness plans, country readiness assessment for health emergencies, missions to the countries, and the most up-to-date country-specific COVID-19 situation analyses.	World Health Organization (https://www.who.int/docs/default-source/coronaviruse/covid-19-sprp-country-status-feb25.pdf?sfvrsn=76f83ed3_1#:~:text=The%20operational%20readiness%20index%20(levels,the%20risk%20of%20COVID%2D19 (accessed on 4 July 2022))
Years of schooling (x_2)	Mean years of total schooling across all education levels.	United Nations Development Program (https://hdr.undp.org/data-center/documentation-and-downloads (accessed on 4 July 2022))
Economic complexity index (x_3)	The index measures how diversified and complex is the export basket of a country.	Atlas of Economic Complexity (https://atlas.cid.harvard.edu/rankings (accessed on 4 July 2022))
Rural population (x_4)	The rural population is calculated using the urban share reported by the United Nations Population Division. There is no universal standard for distinguishing rural from urban areas. That is why, rural population refers to people living in rural areas as defined by national statistical offices.	World Bank (https://data.worldbank.org/indicator/SP.RUR.TOTL.ZS (accessed on 4 July 2022))

**Table 3 vaccines-11-00248-t003:** Descriptive statistics where the main variants of COVID-19 emerged.

Country	Percentage of People Fully Vaccinated	All Countries of the Sample	Years of Schooling	All Countries of the Sample	Crowding	All Countries of the Sample	Percentage of Urban Population	All Countries of the Sample
Brazil (Dseta)	0.000	0.000	8.129	9.928	2.000	0.243	87.073	64.427
Brazil (Gamma)	0.000	0.000	8.129	9.928	2.000	0.243	87.073	64.427
Colombia (Mu)	0.558	2.961	8.863	9.921	0.000	0.262	81.740	64.808
India (Delta and Kappa)	0.000	0.000	6.655	9.941	5.000	0.215	34.926	64.914
Peru (Lambda)	0.000	0.000	9.886	9.911	0.000	0.262	78.297	64.509
Philippines (Zeta)	0.001	2.967	8.969	9.920	1.000	0.252	47.684	65.126
South Africa (Beta)	0.000	0.000	11.373	9.897	0.000	0.262	67.354	64.611
South Africa (Omicron)	26.371	48.992	11.373	9.897	0.000	0.262	67.847	64.938
United Kingdom (Alpha)	0.000	0.000	13.406	9.878	1.000	0.252	83.903	64.457
United Kingdom (Eta)	0.000	0.000	13.406	9.878	1.000	0.252	83.903	64.457
United States of America (Epsilon)	0.000	0.000	13.683	9.876	2.000	0.243	82.664	64.468
United States of America (Iota)	0.013	0.000	13.683	9.876	2.000	0.243	82.664	64.468

**Table 4 vaccines-11-00248-t004:** Logit random effect panel data analysis of the determinants of the appearance of a new relevant COVID-19 variant.

Dependent Variable	Dummy (Takes Value 1 When D > 0)	
Independent Variable	Coefficient (left column)	
	Odd ratios (right column)	
	(1)	
Percentage of people fully vaccinated	−7.142	0.0008
	(3.596) **	(0.003) **
Years of schooling	0.180	1.197
	(0.199)	(0.238)
Crowding	1.974	7.200
	(0.699) ***	(5.036) ***
Urban population	0.057	1.059
	(0.020) ***	(0.021) ***
Intercept	−12.919	0.000
	(4.813) ***	(0.000) ***
Obs	972	
Wald chi2	17.55 ***	
Log pseudo-likelihood	−48.183	

Note: Robust standard errors in parentheses. ***, **, and * indicate significance at 1%, 5%, and 10%, respectively. In the model we also report odds ratios, to be interpreted as the probability of success divided by the probability of failure. For example, the odd ratio for variable 4 (“crowding”) says that among those countries which have crowded cities with more than ten million inhabitants, the probability of the appearance of a new COVID-19 variant is 4.05 times more than the probability that a new variant of COVID-19 appears in a country without crowded cities with more than ten million inhabitants.

**Table 5 vaccines-11-00248-t005:** Marginal probability effects analysis for the determinants of the appearance of a new relevant COVID-19 variant.

Dependent Variable	Dummy (Takes Value 1 When D > 0)
Independent Variable	(1)
Percentage of people fully vaccinated	−0.070
	(0.035) **
Years of schooling	0.002
	(0.002)
Crowding	0.019
	(0.004) ***
Urban population	0.000
	(0.000) ***
Obs	972

Note: Robust standard errors in parentheses. ***, **, and * indicate significance at 1%, 5%, and 10%, respectively.

**Table 6 vaccines-11-00248-t006:** Descriptive Statistics by region.

Region	Operational Readiness Index	Years of Schooling	Economic Complexity Index	Percentage of Rural Population
North America	4.667	12.246	1.085	18.239
Central America	2.875	8.674	−0.0541574	34.069
South America	3	9.494	−0.4158173	20.035
Europe	4.143	12.243	0.9398915	28.452
Africa	2.810	6.671	−0.8807258	49.798
Rich Countries of Asia *	4.5	12.097	0.6558612	13.148
Rest of Asia	3.087	9.010	−0.0106188	45.999
Rest of Countries	4	10.137	−0.6102337	37.845
Countries of the Sample	3.5	9.911	0.1420442	35.181

* Includes Bahrain, Japan, Oman, Republic of Korea, Saudi Arabia, and the United Arab Emirates.

**Table 7 vaccines-11-00248-t007:** Panel Data estimation of the determinants of the vaccination rate process.

Dependent Variable	D|D > 0
Independent Variable	(1)
Operational readiness index	0.040
	(0.015) ***
Years of schooling	0.0032
	(0.001) **
Economic complexity	0.019
	(0.008) ***
Rural population	−0.001
	(0.000) ***
Intercept	0.040
	(0.030)
Obs	972
R-square	0.085
Wald chi2	11.01
Prob > chi2	0.026

Note: Robust standard errors in parentheses. The model has no serial correlation, contemporary correlation, and heteroskedasticity since it is solved using linear regression with correlated panels corrected standard errors. ***, **, and * indicate significance at 1%, 5%, and 10%, respectively.

## Data Availability

The following information can be downloaded at: https://drive.google.com/drive/folders/1GBLaMX9sUNazykJQklMCkaU64GnFJnmP?usp=share_link.

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
