# Peer review of "Lessons from COVID’S Vaccination: External-Internal Frictions and Efficiency"

_vaccines, 2023, doi:10.3390/vaccines11020248_

Round 1

Reviewer 1 Report

The analysis of the determinants of vaccination rate seems fine.  Although the conclusions aren't surprising, the scope of the data makes the results important.  But the other analysis, the emergence of new variants, needs more explanation.  According to table 1, it's the "appearance of a new ... variant before the population reaches a 50% vaccination rate."  That seems to create a negative relationship to the vaccination rate by definition, raising the question of whether the results actually tell us anything.  Also, Table 3 just lists 13 cases, which presumably are the nations in which the variant first emerged.  Is that the dependent variable?  In that case, there should be some discussion of what that means for the statistical analysis (it's essentially like having a very small sample).  Or if the dependent variable is whether the variant eventually took hold in the nation, there should be some discussion of how that is defined (ie, the minimum prevalence to say that it's present).  The general point is that there needs to be more discussion of how the variable is defined, what that means for the analysis, and if possible, some experimentation with alternative definitions. 

A couple of minor points:

1.  The tables include too many significant digits--two or at most three is sufficient. 

2.  The English is generally OK, but there are a number of typos or small mistakes--e. g., "trough a panel" , "surpass largely," "Nuts [presumably Nueces] County in Texas". 

Author Response

We really appreciate your comments and suggestions.

Sincerely,

  1. Aldo Ramírez Zamudio

Department of Economics, Universidad de Lima; aframire@ulima.edu.pe

  1. Elmer Sánchez Dávila

Ph.D. student at the University of Talca - Chile, elmer.sanchez@utal.cl

Reviewer 2 Report

The objective of thismanuscript is to investigate different factors that altering vaccination and its efficiency to achieve herd immunity, please consider the following points before considering to be published in Vaccines:

1. The authors tried to classify the vaccination percentage in term of different countries. However, other than the economic factors, one of the critical factor should be considered. The factor is the “war”. For example Russo-Ukrainian War can be altered the factors of vaccination. Therefore, I think the author should onsider the above factor before making any statement about the relationship between economics and vaccination rate.

2. In line 57 to 67, authors tried to summarize the effective way of vaccine belongs to private laboratories research. However, some findings were revealed that the willingess of accepting vaccination does altered the herd immunity. Please read the paper Wang et al., (2021). Change of Willingness to Accept COVID-19 Vaccine and Reasons of Vaccine Hesitancy of Working People at Different Waves of Local Epidemic in Hong Kong, China: Repeated Cross-Sectional Surveys. Vaccines 9(1), 62. for details.

3. In line 80-83, authors should elaborate the definition of “economic factors”. There are many interpretations and factors that defining “economic factors” including but not limited to interest rates, tax rates, laws, policies, wages, governmental activities, employment, wages, prices/inflation, interest rates, and consumer confidence. As we all know the value of GDP issued by country is an approcximated value. It cannot be reflected totally the real situation of economic status of that country. Therefore, I think author should be investigated the rate of vaccination related to the TRENDS of GDPs in the ten years instead of considering only a short interval year of GDP (i.e. March 2020 to March 2022) because the value of GDP can be altered significantly by the COVID-19 pandemic. The authors should clarify this point before cionducting any analysis on the study. Otherwise, the study is meaningless.

4. In line 96, The statement should be critically reviwed about the sentence…“High vaccination rates and high level of PUBLIC HEALTH education decrease the probability of the ……” . This is because the level of education is not related to the vaccination rate. Please check the paper entitled “Logullo et al., (2008). Factors affecting compliance with the measles vaccination schedule in a Brazilian city. Sao Paulo Med. J. 126 (3).” It implied that authors haven’t had any detail literature review before carrying any analysis. Please redo it accordingly and raised the controverisal questions about the vaccination rate and the promotion of public health education.

5. In line 116 to 188: I cannot find any patterns of literature review. Therefore, please arrange it BASED ON the issue such as vaccination percentage, vaccination willingness, vaccination efficiency, vaccination popularity….etc for example

6. In line 160, is it possible to use the materials in the “FACEBOOK’ as the formal research materials? As we all know the relaibility of FACEBOOK is low and every people can write their view points towards vaccination. Such information cannot be trusted in an attempt to cite in the scientific paper. Fabiana Zollo & Walter Quattrociocchi (2018) wrote Misinformation Spreading on Facebook in the Book named “Complex Spreading Phenomena in Social Systems” (pp 177–196) and it mentioned FACEBOOK may leads to information overload and confirmation bias. Please revised the content of literature review.

7. In line 216,217, 228, the logit panel model has been used for calcualtion. please cite the related formulae and the methodology of estimation and give reasons why such calculation is reliable to your study. The logit model assumes a logistic distribution of errors, and the probit model assumes a normal distributed errors. These models, however, are not practical for cases when there are more than two cases. Therefore, author should provide explaination of using Logit-Panel model.

8. For table 3, the author should explain the rationale of choosing particular country for studies such as Brazil. How about the other countries such as China, Philipines…such countries were also attacked by the same type of pandemic such as Delta, Omicron main strains of COVID-19 emerged. Besides, the data for 0.000 is meaningless. Please explain the data.

9. For the explaination, I think the herd immunity of the particular city can be achieved not only by the vaccination rate, please also explain the data according to the contextual conditions of the cities as follows: High density, Tall buildings, Concentrated in a small area, Mixed land uses, High connectivity , Easy access, A growing and ageing population, ecological footprint…etc.

10. In line 327, how the author define “rural populations”? There are different countries have different interpretations on the term. Some countries such as India are mainly migrated population and does effect the data significantly. Please refer to the publication for your reference and evaluate the limitation of the current findings. R Suresh, J James, B RSj. 2020. Migrant workers at crossroads–The COVID-19 pandemic and the migrant experience in India. Social Work in Public Health 35, Pages 633-643.

Author Response

(The authors gave the same response as above.)
